# AdFlush: A Real-World Deployable Machine Learning Solution for Effective Advertisement and Web Tracker Prevention

## ABSTRACT

Ad blocking and web tracking prevention tools are widely used, but traditional filter list-based methods struggle to cope with web content manipulation. Machine learning-based approaches have been proposed to address these limitations, but they have primarily focused on improving detection accuracy at the expense of practical considerations such as deployment overhead. In this paper, we present AdFlush, a lightweight machine learning model for ad blocking and web tracking prevention that is practically designed for the Chrome browser. To develop AdFlush, we first evaluated the effectiveness of 883 features, including 350 existing and 533 new features, and ultimately identified 27 key features that achieve optimal detection performance. We then evaluated AdFlush using a dataset of 10,000 real-world websites, achieving an F1 score of 0.98, which outperforms state-of-the-art models such as AdGraph (F1 score: 0.93), WebGraph (F1 score: 0.90), and WTAgraph (F1 score: 0.84). Importantly, AdFlush also exhibits a significantly reduced computational footprint, requiring 56% less CPU and 80% less memory than AdGraph. We also evaluated the robustness of AdFlush against adversarial manipulation, such as URL manipulation and JavaScript obfuscation. Our experimental results show that AdFlush exhibits superior robustness with F1 scores of 0.89–0.98, outperforming AdGraph and WebGraph, which achieved F1 scores of 0.81–0.87 against adversarial samples. To demonstrate the real-world applicability of AdFlush, we have implemented it as a Chrome browser extension and made it publicly available. We also conducted a six-month longitudinal study, which showed that AdFlush maintained a high F1 score above 0.97 without retraining, demonstrating its effectiveness. Additionally, AdFlush detected 642 URLs across 108 domains that were missed by commercial filter lists, which we reported to filter list providers.

**ACM Reference Format:**
Anonymous Author(s). 2024. AdFlush: A Real-World Deployable Machine Learning Solution for Effective Advertisement and Web Tracker Prevention. In *Proceedings of International World Wide Web Conference 2024 (WWW '24)*. ACM, Singapore , 12 pages. https://doi.org/XXXXXXX.XXXXXXX

## 1 INTRODUCTION

Many internet users are concerned about web advertisements (ads) and trackers that display unsolicited content or collect their personal information without explicit consent. To address these issues, several tools are available, AdBlock Plus, uBlock Origin, and

Ghostery. These tools use filter lists, such as EasyList [9] and EasyPrivacy [10], which contain rules for determining whether HTTP requests are associated with ads or trackers. However, manual maintenance of these filter lists requires significant human effort, and they are prone to false-positive and false-negative errors [1]. The rules primarily rely on specific patterns, which renders them ineffective against evasive or manipulated content.

Machine learning (ML) detection techniques [19, 21, 37, 43] have recently been investigated to address the limitations of traditional filter list-based methodologies. AdGraph [21] and WTAGraph [43] construct graphs to extract significant features based on HTTP traffic. Furthermore, WebGraph [37] utilizes additional behavioral features to build more robust detection models against adversarial evasion techniques. However, these studies have mainly focused on developing highly accurate detection models by incorporating new features to identify ads and trackers. While ML models show great promise in identifying them, they often rely on extensive feature sets. This can lead to significant execution time and memory overhead for feature extraction, model training, and inference. This may not be suitable for browser settings on a typical desktop machine due to limited computational resources.

To deploy ML detection models in standard web browsers, we need to identify a minimal yet effective set of features that can be efficiently extracted without compromising accuracy. This involves studying the features used by existing models, analyzing their relevance, and refining the feature set based on considerations such as computational complexity and feasibility in a browser environment.

To achieve this goal, we carefully examined all potential features from prior models and analyzed their impact on detection accuracy and feasibility in web browser settings. Based on our feature engineering, we selected an optimal set of features for detecting advertisements and web trackers. We then used automated machine learning (AutoML) [27] on these features to identify the most efficient machine model, AdFlush.

To demonstrate the effectiveness of AdFlush, we evaluate the performance of AdFlush against state-of-the-art models (AdGraph [21], WebGraph [37], and WTAGraph [43]) using the following performance evaluation metrics: F1 score, training time, inference time, memory usage, and CPU usage. AdFlush achieves an F1 score of 0.98, outperforming AdGraph (F1 score: 0.93), WebGraph (F1 score: 0.90), and WTAGraph (F1 score: 0.84). The average inference time for AdFlush over 830K traffic samples is 2.3 seconds, considerably faster than AdGraph's 17.4 seconds. AdFlush has demonstrated excellent computational efficiency by exhibiting 56% lower CPU usage and 80% less memory consumption than AdGraph. Regarding its robustness against adversarial attacks such as URL manipulation and JavaScript obfuscation, AdFlush significantly outperforms AdGraph and WebGraph's performance.

Unlike previous research, we consider the feasibility of the features within real-world browser settings, demonstrating that AdFlush can be deployed as a web browser extension without causing

significant CPU or memory overheads. Furthermore, AdFlush does not transmit `User-Agent` values or cookies to third parties, upholding a strong commitment to user privacy. Our main contributions can be summarized as follows:

- We propose a new set of 27 features for detecting ads and trackers in real-world browser extension environments. The feature set was carefully selected from a pool of 883 features, including 350 existing features from state-of-the-art methods [21, 37, 43] and 533 new JavaScript features that are robust to adversarial samples with JavaScript obfuscation.
- We developed AdFlush, a lightweight yet highly accurate model for detecting advertisements and web trackers using AutoML [27]. To demonstrate its practicality, we implemented AdFlush as a Chrome extension. The source code and dataset for AdFlush are available at *https://anonymous.4open.science/r/AdFlush-4EF0* We have also made a comprehensive video demo available at *https://youtu.be/dzdfqpiCjKg* demonstrating AdFlush's practicality in real-world web browser settings.
- We evaluated the long-term effectiveness and adversarial robustness of AdFlush. In a six-month longitudinal study, AdFlush maintained a high F1 score above 0.97 without retraining, demonstrating its effectiveness. Commercial filter lists lagged by an average of 80 days in updating ads and trackers, while AdFlush detected 642 URLs across 108 domains that were missed by filter lists. Benchmarked against state-of-the-art models using three adversarial test cases (URL manipulation, JavaScript obfuscation, and ML-based adversarial sample generation), AdFlush consistently outperformed other models.

## 2 RELATED WORK

### 2.1 Risks of Advertisements and Trackers

In today's digital marketing environment, web advertising and tracking practices are essential for providing tailored user experiences, but they also raise privacy and security concerns [28].

Web trackers monitor users' online activities, such as visited sites, duration, clicked links, and location. This data helps create detailed profiles for targeted advertising [4]. While some find these ads useful, others see them as privacy invasions. Many firms distribute this data without user consent, raising data breach risks [23]. This heightens the threat of cyberattacks and identity theft for users.

Web ads and tracking can expand the attack surface of users. One such threat is *malvertising*, where malicious actors exploit advertisements to distribute malware or exploit system vulnerabilities [5]. These threats can be present on any website, even those generally perceived as safe, posing a danger to users [17]. Other security concerns originate from web tracking technologies like cookies and browser fingerprinting. Cookies, small files stored on a user's device, could be exploited for harmful purposes. Browser fingerprinting, which identifies users based on their browser's statistical properties, can be manipulated by malicious actors to conduct targeted attacks [12, 30].

### 2.2 Filter List-Based Approaches

Many web users rely on browser extensions such as AdBlock Plus, uBlock Origin [18], Privacy Badger [33], and Disconnect [8] to protect themselves from intrusive ads and privacy-invading web trackers. Recognizing the importance of these ad blockers, several web browsers, including Firefox [14], Ghostery, and Brave [3], now offer built-in ad-blocking solutions. This demonstrates the importance of these extensions in the browsing environment.

Ad blocking solutions primarily rely on widely recognized filter lists such as EasyList [9], EasyPrivacy [10], and Fanboy's List [13]. These lists are manually curated lists of websites and scripts to block. While filter lists have been instrumental in the crowdsourced effort to block unnecessary and potentially harmful content, they have several shortcomings [20]. Filter lists can be labor-intensive and challenging to scale, particularly for a large number of websites. Additionally, the crowdsourcing model used to develop filter lists can result in regional bias, as regions with fewer contributors may have fewer ad-blocking rules on the list [38]. Finally, filter lists struggle to keep up with the growing trend of script obfuscation used by web trackers [11].

### 2.3 ML-Based Approaches

To overcome the limitations of traditional methods that rely on manually created filter lists, various techniques have been proposed for generating filter lists in an automated manner. For instance, Bhagavatula *et al.*'s model [2] streamlines the process of creating and maintaining filter lists. Recently, AutoFR [26] introduced an efficient filter list generation method using reinforcement learning. However, automated filter rule generation is inherently unsuitable for real-time ads and trackers detection.

Recent graph-based ML approaches, such as AdGraph [21], PageGraph [38], WTAGraph [43], and WebGraph [37], utilize features from ads and trackers' interactions across HTML, HTTP, and JavaScript layers to train models for detection. PageGraph extends AdGraph's graph representation by refining event attribution and capturing more behaviors, but both overlook HTTP redirects and request header information. WebGraph [37] addresses this by adding features that trace information flow within a browser, including HTTP redirects and request headers. Given the natural graph structure of HTTP network traffic, WTAGraph leverages a graph neural network (GNN) model for better detection accuracy. However, both WebGraph and WTAGraph operate offline only in practice due to performance overheads, which subjects them to challenges similar to traditional filter lists. Notably, WebGraph's use of over 3,000 features without GPU acceleration results in lengthy inference times, making real-time desktop operation impractical for average users.

Another approach involves utilizing computer vision techniques to detect ad images. Storey *et al.* [39] introduced Ad Highlighter, the inaugural perceptual ad-blocker that identifies ad disclosures by merging web-filtering rules with computer vision methods. Due to the enhanced robustness of perceptual techniques, many contemporary ad-blockers have integrated similar concepts. For instance, Adblock Plus has adopted image-matching filters. Percival [7] employs CNN classification models for advertising image detection. However, challenges arise when ads are placed outside of iframes, use transparent logos, or are fragmented into smaller images. These tactics hinder the efficacy of computer vision in detecting ads [40].

ML-based approaches have potential for ad-blocking tools, but they have not been widely adopted yet due to performance and reliability issues. Performance-wise, the overheads associated with

feature instrumentation and executing ML pipelines in real-time can offset the benefits of ad blocking [26]. Reliability issues arise from doubts about the robustness of ML models in real-world settings [7]. Additionally, recent graph-based ML models typically aggregate features only after webpages have fully loaded, potentially allowing ads and trackers to execute before feature extraction. Evaluations often neglect practical considerations, including the model's impact on webpage load time, CPU and memory overhead, and robustness against adversarial attacks.

Therefore, we propose a practical ML-based solution designed for seamless integration as a browser extension to address these limitations. We aim to achieve high detection accuracy using only practical features supported by commercial web browsers.

## 3 FEATURE ANALYSIS

This section comprehensively analyzes 883 potential features, including 350 used by existing ML-based methods and 533 new features that are robust to adversarial samples with JavaScript obfuscation. We assess the feasibility and contribution of each feature to overall detection efficacy, identifying key features that can be practically implemented in web browsers.

### 3.1 Data Collection

To collect a comprehensive dataset of real-world websites, we sampled the top 10,000 websites ranked by Tranco [32]. This was because Alexa, previously used in studies for AdGraph [21] and Web-Graph [37], no longer provides an accessible or updated ranking service. We also followed the approach of WTAGraph [43], which used the top 10,000 sites based on Tranco's ranking.

We set a 60-second timeout for each website to account for the variability in complexity and structure of different websites. Our data collection process resulted in an average crawl time of 32.58 seconds, with a standard deviation (SD) of 11.26 seconds, indicating that a 60-second timeout is appropriate for most sites. With seven web crawler instances running in parallel for 11 hours using OpenWPM[11], we generated a total of 830,160 requests. We split the dataset into an 8:2 ratio for training and testing. This ensures that the model is trained on a representative sample of the data and evaluated on a held-out set that it has not seen before.

### 3.2 Candidate Features

We thoroughly analyzed the potential features present in our gathered dataset, integrating all features leveraged by AdGraph [21], WebGraph [37], and WTAGraph [43]. This resulted in a total of 3,323 distinct features, with some overlap.

AdGraph focuses on monitoring DOM tree alterations from JavaScript executions or HTTP requests. The resulting changes are formulated into a graph, from which 28 features are derived. These features encompass both the graph's structural attributes and content-based features, such as the attributes of the request URL.

WebGraph builds upon AdGraph by representing a broader spectrum of components in its graph structure, including HTML, network, JavaScript, and storage components. Drawing from this graph, WebGraph uses 59 features that encapsulate both structural attributes and advanced dataflow characteristics, with special emphasis on cookie access and information sharing between nodes.

WTAGraph constructs an attribute homogeneous multi-graph (AHMG) from the visited web pages. This complex network then translates into a feature vector that includes, among others, character strings, specific JavaScript APIs, and HTTP request metrics. In AdFlush, we condensed WTAGraph's vector space of 3,200 URL character representations into 30 features for FQDNs and 200 for request URLs using continuous bag-of-words features. This approach strikes a balance between computational feasibility and robustness against overfitting.

Moreover, we introduced 533 new features to enhance AdFlush's robustness against JavaScript obfuscation techniques. Traditional features were insufficiently resilient against adversarial samples employing JavaScript obfuscation. These new features are derived from requested JavaScript source code or embedded HTML scripts, including $n$-gram frequencies parsed from the abstract syntax tree (AST), the structure of the AST, and various script-based metrics.

We present a comprehensive feature list by combining 350 features from three existing models [21, 37, 43] with 533 additional JavaScript-specific features (detailed in Appendix A). Some features, such as URL character embeddings, are combined into a single feature label for clarity. We used the code provided in the research to extract features whenever possible. Otherwise, we implemented the code from scratch, especially for proprietary features or when the source code from the baseline models was unavailable.

### 3.3 Feature Categorization

As presented in Appendix A, we categorize our features into four groups: JavaScript, URL, HTTP header, and Graph as follows:

- **JavaScript** features are derived from the HTML structure of a web page, focusing on functions within <script> tags and JavaScript code, such as fetch() or XMLHttpRequest. They also monitor specific API usage, revealing downstream behavior of advertising and tracking actions. For example, eval() can trigger hidden <script> tags to collect user tracking data, while code tracking significant user interaction often pertains to advertising.
- **URL** features represent values from source and request URLs. Keywords in the request URL or its length can provide insights into the request's purpose. Though ad and tracker services use various URL manipulation techniques, embedding URLs can effectively counter these methods.
- **HTTP header** features are derived from values in HTTP request headers. These encompass content policy type, User-Agent strings, referrer data, and other metadata embedded in the HTTP request header. These features are crucial as they offer context about a request or response, often containing user-identifying information or insights about the request's nature.
- **Graph** features illustrate interactions between different graph nodes, representing various web page elements, with edges representing interactions between them. Potentially important features include the distance between nodes, node centrality, and the number of requests or redirects involving a node. These features have recently received increased research attention due to their potential to highlight the complex dynamics of various components on web pages.

In our categorization, AdGraph primarily utilizes URL and Graph features, incorporating 1, 11, 1, and 15 features from each category, respectively. WebGraph emphasizes Graph features, using 7 JavaScript and 52 Graph features. WTAGraph primarily uses 32 JavaScript, 3,200 URL, and 34 HTTP header features to construct a GNN structure with these features.[1]

We evaluated the feasibility of integrating each feature into standard web browsers, according to the following rules.

- **Rule 1 (Avoiding loading additional resources)**: Feature extraction should not require executing JavaScript, WebAssembly, or loading extra browser elements. As AdFlush aims to intercept ads and trackers before they load, it should depend solely on features that avoid inspecting interactions that could introduce the user to undesired ads or trackers.
- **Rule 2 (Avoiding using multiple requests)**: Feature values should be derived from a single web page request to ensure swift detection and real-time analysis without excessive memory or storage use. Nested queries on various requests demand user history and significant computation.

Under these rules, most Graph features are infeasible because they violate **Rule 1** or **Rule 2**. For example, the *number_of_set_then _get_storage* feature observes data sharing nodes in order, yet requires JavaScript execution with timestamps and violates **Rule 1**. The more complex *indirect_all_average_degree_connectivity* feature cannot be computed without fetching all nodes and edges in a graph, violating **Rule 2** because it requires access to the user's site visit history to access data about all connected nodes, raising potential privacy concerns.

The only Graph feature efficiently computed with a single request is *is_parent_script*. This feature determines whether the initiator of the request is a JavaScript resource. Therefore, *is_parent_script* is the only Graph feature we consider feasible to implement in real-world browser settings. As a result, we excluded 52 Graph features out of the 350 features from our candidate feature set.

## 3.4 Key Feature Identification

To identify the most crucial web tracking and ad detection features, we evaluated the relevance, impact, and association with ad and web tracker detection of 298 established state-of-the-art features and 533 newly proposed features. We identified key features from each group separately, including the 533 newly proposed features, even though they exhibited lower importance in normal samples. These features improve resilience against JavaScript obfuscation tactics, which complicates the task for ML-based detectors in distinguishing between malicious and benign JavaScript code [34].

We first examined the correlation between each feature and the labels using the point-biserial correlation coefficient [24]. We discarded 17 and 178 features from the existing and new sets, respectively, with a *p*-value exceeding 0.1, indicating that these features have an insignificant correlation with the labels and are, therefore, unlikely to be relevant to the model.

We next curated the key feature sets for AdFlush from the remaining 281 existing and 355 new features. We used recursive feature elimination with cross-validation (RFECV) [16] to identify the most influential features. RFECV works by iteratively removing features with the lowest importance scores and evaluating the model's performance based on the remaining features. Figure 1 shows the results of the RFECV process, with the x-axis denoting the number of features and the y-axis indicating the cross-validation accuracy. The scores for the new features are lower than those for the existing features. For the existing features, accuracy increases as we remove features until we retain 47, after which it decreases. For the new features, optimal results are achieved with 11 features. Therefore, we considered a set of 58 features, combining the 47 and 11 features from the two groups.

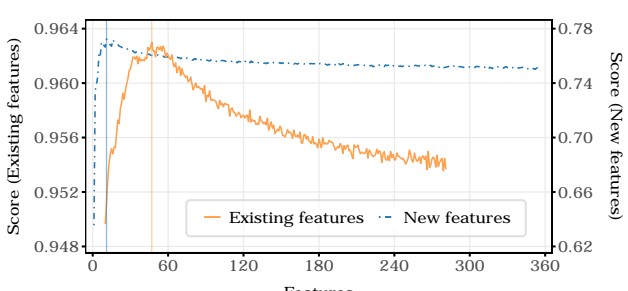

**Figure 1: RFECV scores against feature count.**

To further streamline the number of features, we aimed to select only those among the remaining 58 features that exhibited minimal correlation with each other. Thus, we analyzed the correlation for all possible pairs within these 58 features. We employed Pearson's coefficient for pairs with continuous values and used Spearman's correlation otherwise. If the correlation *p*-value between any two features was below 0.05, we discarded the less significant feature based on RFECV scores. This process resulted in a final set of 27 features. The comprehensive list of features in AdFlush is detailed in Appendix B.

We used a random forest to assess the importance of each feature in the final 27 features. We distinguished ads, trackers, and regular resources using three features: the number of storage elements retrieved by the resource, the URL length, and the number of cookie values retrieved. These features are crucial for identifying ads and trackers because they often store user identifiers in storage elements, have long request URLs, and use more cookies for tasks such as cookie synchronization. We observed a variation in the distribution of these features in our dataset, with ads and trackers typically having higher average values for all three features (see Figure 2). For example, ad and tracker resources sometimes accessed storage elements over 300 times, while regular resources rarely did so. While some features might have a minor individual impact, their collective use produces a robust indicator for differentiating ads from trackers. We found 61,285 ads and trackers that receive cookie values, access storage elements, and have a URL length of over 200 characters, in contrast to only 6,793 instances for regular resources. This suggests that these features can be used to distinguish between ad/tracker and regular resources.

---

[1]The feature counts mentioned here do not align with the number of ● symbols in Appendix A. This discrepancy occurs because we group together features computed using similar methods in the table.

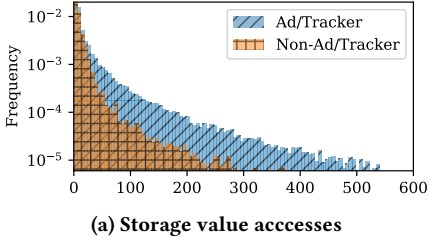
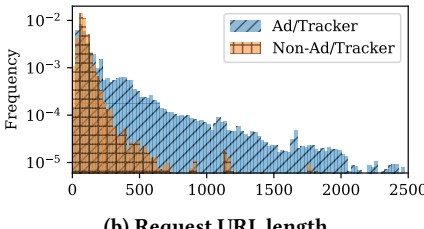
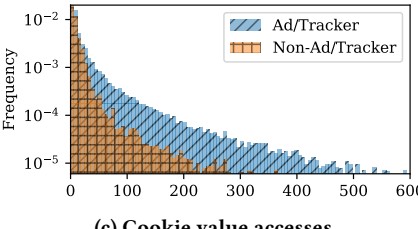

**(a) Storage value acccesses**          **(b) Request URL length**          **(c) Cookie value accesses**

**Figure 2: Distribution of three features for ad/tracker versus non-ad/tracker resources (normalized, y-axis in log scale).**

Table 1 shows the relative importance of the top five newly introduced JavaScript features in terms of information gain. The most important feature is the 3-gram pattern *(Statement, CatchClause, Statement)*, which is typically represented by try-catch blocks. The other top features, *(Expression, Identifier, Identifier)* and *(Expression, Expression, Statement)*, are less distinctive. However, they still contribute to the model's performance, with information gains of 1.27% (± 0.19) and 1.26% (± 0.20), ranking 18th and 19th in importance, respectively. Additionally, obfuscation-related features, such as *Identifier* lengths in ASTs and character counts in code lines, would also enhance performance.

**Table 1: Top-5 important JavaScript features and information gain within optimized 27 features (averaged over 10 folds).**

| Feature | Information Gain (%) |
| --- | --- |
| *(Statement, CatchClause, Statement)* | 5.42 ± 0.54 |
| Average of *Identifier* length | 2.24 ± 0.32 |
| Average of character per line | 1.32 ± 0.22 |
| *(Expression, Identifier, Identifier)* | 1.27 ± 0.19 |
| *(Expression, Expression, Statement)* | 1.26 ± 0.20 |

## 4 IMPLEMENTATION OF ADFLUSH

We aim to identify the most effective features for improving the performance of AdFlush while minimizing its complexity. To achieve this goal, we carefully selected 27 features with the highest importance scores. We then applied a dimensionality reduction technique to this compact feature set, further simplifying the model and improving its predictive power by preventing overfitting.

To find the most efficient models for these reduced feature spaces, we used H2O AutoML [27]. This tool provides a comprehensive search and selection platform, allowing us to explore different models to identify those that provide the highest performance. Once we had identified the best-performing models, we optimized them to be more efficient and resource-friendly.

We integrate our model into a fully functional web browser extension for Chrome. This extension captures features in real-time and promptly blocks ads and trackers detected by the model, ensuring a secure browsing experience (see Figure 3).

### 4.1 Dataset

We aggregated eight popular filter lists (see Appendix C) to label our dataset accurately and distinguish between the presence or absence of ads and trackers. Our labeling method mirrors that used in existing studies [20, 37], ensuring consistency and comparability. After labeling, we divided the data into training and test sets in an 8:2 ratio.

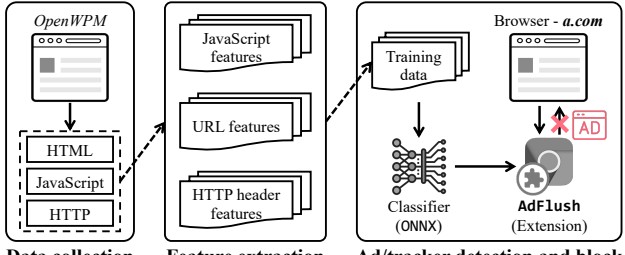

**Figure 3: Overview of AdFlush.**

This substantial labeled dataset was then used to train and test the detection model and to compare it with state-of-the-art solutions.

### 4.2 Model Selection and Training

We used H2O AutoML [27], an automated machine learning platform, to find the best model with our feature set. AutoML facilitates the search for high-performing models and the generation of optimized feed-forward artificial neural network models. We analyzed the training data for each feature selection, identified the top 10 models based on F1 scores, and selected the best-performing model using 5-fold cross-validation. After this rigorous evaluation, a gradient boosting machine (GBM) [36] was selected as the best-performing model. For AdFlush, we selected 27 features based on their importance scores and correlation (see Section 3.4). However, our analysis found significant correlations among these features. We could use dimensionality reduction techniques to streamline the feature set and model while preserving pivotal information. We considered two dimensionality reduction techniques: Principal Component Analysis (PCA) [25] and Uniform Manifold Approximation and Projection (UMAP) [29]. While PCA is a robust technique, it can overlook subtle nuances, particularly in datasets with non-linear characteristics. To address this, we also considered UMAP. We condensed the 27 features with PCA to 8, and with UMAP, we reduced them to 2. These results are consistent with our correlation analysis, suggesting that the 27 features may contain redundant data, with some being highly interrelated.

We trained the GBM with each of these three feature sets to determine the optimal feature set and evaluated the performance of AdFlush on the test set (see Table 2).

Although reducing AdFlush's feature set to two features using UMAP achieved the highest F1 score of 0.99, we chose to retain the original 27 features because using UMAP would have complicated result interpretation and increased computational and memory

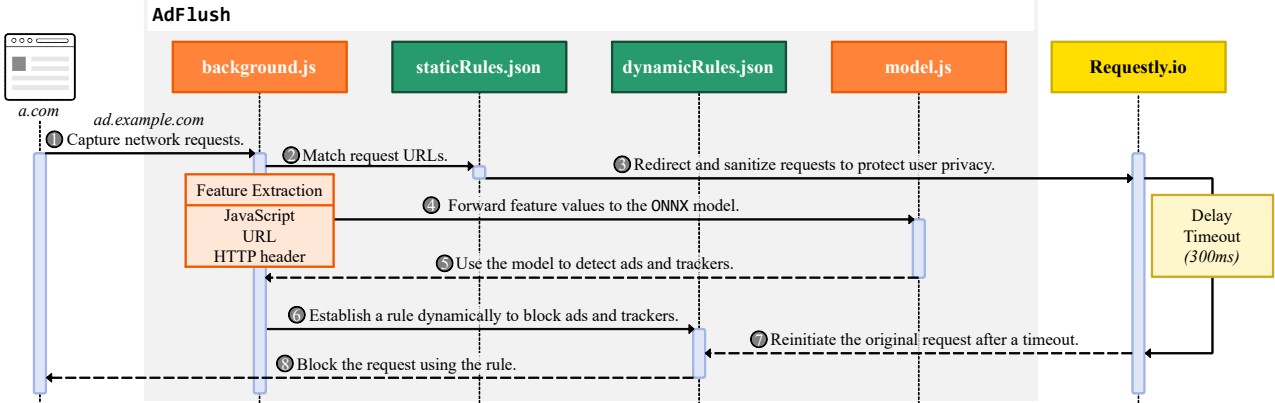

**Figure 4: Sequence diagram of AdFlush in browser extension.**

**Table 2: Performance of AdFlush with each feature set: Original (27), PCA (8), and UMAP (2).**

| Feature Set | Acc. | Prec. | Recall | $F_1$ |
|---|---|---|---|---|
| **AdFlush** Original (27) | **0.98** | **0.99** | 0.97 | 0.98 |
| **AdFlush** PCA (8) | 0.96 | 0.96 | 0.93 | 0.95 |
| **AdFlush** UMAP (2) | **0.98** | **0.99** | **0.98** | **0.99** |

overhead due to the additional processing steps. Since the performance improvement was marginal (0.99 vs. 0.98), we prioritized interpretability and efficiency and used the original features.

### 4.3 Browser Extension Implementation

Google's recent adoption of Manifest V3 [6] restricts Chrome extensions from directly manipulating user HTTP requests. Therefore, extensions must define rules or conditions for blocking requests rather than controlling them. To comply with this new standard, we implemented AdFlush, a Chrome browser extension that uses a GBM model to generate rules on the fly to block ads and web trackers.

Our implementation approach adheres to the guidelines of Manifest V3, which emphasize the importance of performing model inference in real time to generate rules. We used the libraries onnxmltools and ONNX-simplifier to convert our GBM model into the ONNX format, which enables efficient execution through a web assembly backend. Our extension for the Chrome web browser uses this model to detect ads and web trackers from web page requests in real-time.

Figure 4 illustrates the detailed workflow: ① AdFlush captures a network request from the user's active web page. ② A static rule is applied for each matched request. ③ The matched request is redirected to the Requestly delay API via declarativeNetRequest. Before this, AdFlush removes user cookies and appends a dummy User-Agent to safeguard user privacy. In this step, AdFlush extracts features like content policy type, domain party, URL advertising keyword presence, stored value count, and $n$-gram frequencies. It references a pre-trained vector dictionary in JSON format for URL word embedding features. ④ The service worker forwards these values to the loaded ONNX model in an off-screen tab. ⑤ The model predicts and notifies the service worker based on these values. ⑥ If identified as an ad or tracker URL, a dynamic rule is established in declarativeNetRequest; otherwise, benign requests are allowed. ⑦ After the Requestly timeout, the original request is reinitiated. ⑧ Requests violating user privacy (*i.e.*, ad or tracker URLs) are blocked.

Our current implementation of AdFlush relies on the deprecated onBeforeSendHeaders function from the webRequest package. The latest declarativeNetRequest package restricts the reading of raw network traffic, which poses challenges for specific machine learning models that depend on this capability. While Google's Manifest V3 disallows the manipulation of raw requests using webRequest, it still allows for reading. To facilitate this functionality, we used an older version of the declarativeNetRequest package, which enabled us to deploy our AdFlush extension on the latest Chrome version (version 118).

AdFlush's source code and dataset are available at *https://anonymous.4open.science/r/AdFlush-4EF0*. A comprehensive video demonstration is also available at *https://youtu.be/dzdfqpiCjKg*

### 5 EVALUATION

We conducted a comprehensive evaluation to demonstrate the efficacy and efficiency of AdFlush. This included comparing AdFlush to state-of-the-art models, assessing its robustness to adversarial inputs, and measuring the runtime overhead of the browser extension.

### 5.1 Comparison with State-of-the-Art Models

To evaluate AdFlush's detection capability, we compared it with state-of-the-art ad and tracker detection systems (AdGraph [21], WebGraph [37], and WTAGraph [43]). Additionally, we compared it with various machine learning models, including RandomForest, SkopeRules, XGBoost, CatBoost, and LightGBM, using the 27 features identified in AdFlush. Since the source code for feature extraction was not publicly available for WTAGraph, we re-implemented the feature extraction module based on the details provided in the paper. We excluded PageGraph [38] in the comparison because it relies on images within web pages to detect ads. To verify the importance of these features, we also built GBM models in AdFlush using the features from AdGraph and WebGraph, respectively, and

analyzed their performance. All models were evaluated on a dataset comprising 20% of our collected data.

Table 3 shows that AdFlush outperforms other solutions across all metrics, including accuracy, precision, recall, and F1 score. Notably, AdFlush achieved an F1 score of 0.98, significantly higher than the second-best, AdGraph, which scored 0.93. We also found that the 27 features used in AdFlush are highly effective, as evidenced by the RandomForest, XGBoost, CatBoost, and LightGBM models also achieving F1 scores exceeding 0.9. However, the Skope-Rules model showed poor performance with an F1 score of 0.77. When we replaced the 27 features identified in AdFlush with the 28 features from AdGraph and the 59 features from WebGraph, the performance produced F1 scores of 0.91 and 0.86, respectively, demonstrating the importance of our chosen 27 features for achieving high detection accuracy.

**Table 3: Comparison of model performance.**

| Model | # features | Acc. | Prec. | Recall | $F_1$ |
|---|---|---|---|---|---|
| AdGraph [21] | 28 | 0.94 | 0.92 | 0.94 | 0.93 |
| WebGraph [37] | 59 | 0.92 | 0.91 | 0.90 | 0.90 |
| WTAGraph [43] | 3,266 | 0.90 | 0.85 | 0.83 | 0.84 |
| RandomForest | 27 | 0.98 | 0.98 | 0.96 | 0.97 |
| SkopeRules | 27 | 0.83 | 0.87 | 0.68 | 0.77 |
| XGBoost | 27 | 0.96 | 0.97 | 0.94 | 0.95 |
| CatBoost | 27 | 0.95 | 0.95 | 0.92 | 0.94 |
| LightGBM | 27 | 0.94 | 0.94 | 0.91 | 0.92 |
| **AdFlush** | 28 (AdGraph) | 0.93 | 0.90 | 0.92 | 0.91 |
| **AdFlush** | 59 (WebGraph) | 0.89 | 0.85 | 0.87 | 0.86 |
| **AdFlush** | 27 | **0.99** | **0.99** | **0.98** | **0.98** |

We evaluated AdFlush's performance across 14 HTTP request types (see Appendix D). Remarkably, AdFlush achieved an F1 score over 0.93 for all 14 types, demonstrating its effectiveness across all HTTP request types. AdFlush also performed efficiently for types with fewer than 1,000 requests.

## 5.2 Longitudinal Study

Websites constantly evolve, with features frequently updated due to redesigns, new content additions, or algorithm changes targeting ad serving or user tracking. As a result, ML-based solutions typically require periodic retraining on fresh datasets to maintain their performance. However, we show that AdFlush maintains its effectiveness over extended durations without retraining.

We conducted a longitudinal study to evaluate AdFlush's persistent robustness and reliability, especially compared to frequently updated filter lists. We collected eight popular filter lists, updated every four days for six months, following EasyList's update schedule from April 2, 2023, to September 17, 2023. After training AdFlush on the initial training dataset, we assessed its performance against the evolving filter lists, simulating a real-world post-launch scenario.

The performance trajectory of our detection model on the test dataset is illustrated in Figure 5. Impressively, without retraining, AdFlush exhibited robust detection capabilities throughout, maintaining an F1 score above 0.9789. Notably, the F1 score increased until April 10 and then decreased by a negligible 0.0064 over the entire period, demonstrating AdFlush's enduring consistency over extended intervals.

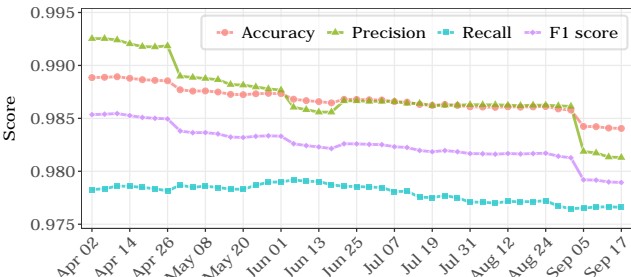

**Figure 5: Longitudinal study from April to September 2023.**

AdFlush's precision gradually declined over time, but its recall rate remained high. This suggests that AdFlush may have increased false positives, but it remained effective at detecting most ads and trackers, even when trained on an older dataset. Appendix E provides a detailed analysis of false negative samples of filter lists.

Commercial filter lists added newly identified ads and trackers with a significant delay (mean = 80 days, SD = 29.89 days) compared to AdFlush. Additionally, AdFlush can detect ads and trackers that are missed by even the latest filter lists, as of October 11, 2023. We manually verified 642 such URLs from 108 unique domains and reported them to EasyList, EasyPrivacy, uBlock Origin, and uBlock Privacy. This demonstrates AdFlush's potential as a practical tool for updating filter lists.

## 5.3 Robustness against Adversarial Samples

ML models require rigorous validation to ensure their resilience to unseen data and feature-targeted attacks. We evaluated AdFlush's robustness against AdGraph [21] and WebGraph [37] under three mutation scenarios, excluding WTAGraph [43] due to its larger feature set and lower effectiveness than WebGraph. We used attack success rate (ASR) and AUROC as experiment metrics. Our experiments are as follows:

**Case 1: URL Manipulation.** Attackers with access to filter lists and a broad understanding of URL patterns used in ad and tracker detection can manipulate URLs to bypass ad blockers. They can change the domain or subdomain name, hide ad and tracker keywords within the query string, or combine these techniques. For example, an attacker might manipulate the URL of an ad from https://secure.adnxs.com/seg?ad=22932261&t=1 to https://UJu0xD.OBGFCuxmU.com/seg?tk=22932261&62Jo=M1lrD0yUI. In our evaluation, we applied all possible combinations of URL manipulation to our test dataset.

**Case 2: JavaScript Obfuscation.** To mimic the tactics of ad and tracker writers attempting to evade detection, we obfuscated JavaScript code using three widely recognized tools: JavaScript-Obfuscator [22], gnirts [15], and Wobfuscator [35]. gnirts is particularly effective at mangling string literals beyond simple hexadecimal string escapes, while JavaScript-Obfuscator offers various code transformations, such as variable renaming and control flow flattening. Wobfuscator replaces parts of JavaScript code with WebAssembly modules that retain the same function. Using these tools, we obfuscated 13,695, 15,386, and 13,311 samples from our test dataset, respectively. We then evaluated the resilience of our model against JavaScript obfuscation using these subsets.

                                                      

**Case 3: CTGAN-Based Manipulation.** We generated adversarial samples using the conditional tabular GAN (CTGAN) [42]. CTGAN is designed to emulate the diversity of real-world data and has shown increased resilience and adaptability across various datasets. Using CTGAN, we generated adversarial versions of our test dataset, tailored to the features of our target models. We first divided our test dataset into an 8:2 ratio for training and testing. Once trained, CTGAN created adversarial samples indistinguishable from real-world data but also designed to evade detection by the tested models.

Table 4 presents the robustness of AdFlush against adversarial samples compared to AdGraph and WebGraph. We excluded Web-Graph when manipulating URL features, as it does not use them, and excluded AdGraph during JavaScript manipulations due to its reliance on a single constant JavaScript feature. Overall, AdFlush consistently demonstrated the lowest ASR and highest F1 score across all adversarial sample scenarios compared to the state-of-the-art models. Specifically, AdGraph's F1 score dropped significantly to 0.81 against CTGAN-based manipulations, while AdFlush maintained a high F1 score of 0.98. WebGraph's F1 score decreased to 0.81 when JavaScript was obfuscated, while AdFlush consistently held an F1 score above 0.89.

**Table 4: Robustness against adversarial samples.**

| Method | Model | Acc. | Prec. | Recall | $F_1$ | ASR | AUROC |
|---|---|---|---|---|---|---|---|
| URL manipulation | AdGraph [21] | 0.89 | 0.85 | 0.86 | 0.85 | 0.11 | 0.95 |
| | **AdFlush** | **0.98** | **0.97** | **0.96** | **0.97** | **0.02** | **0.98** |
| JavaScript-obfuscator | WebGraph [37] | 0.86 | 0.87 | 0.76 | 0.81 | 0.14 | 0.92 |
| | **AdFlush** | **0.95** | **0.96** | **0.91** | **0.94** | **0.05** | **0.94** |
| gnirts | WebGraph [37] | 0.87 | 0.88 | 0.75 | 0.81 | 0.13 | 0.92 |
| | **AdFlush** | **0.92** | **0.99** | **0.81** | **0.89** | **0.08** | 0.90 |
| Wobfuscator | WebGraph [37] | 0.86 | 0.87 | 0.76 | 0.81 | 0.14 | 0.92 |
| | **AdFlush** | **0.95** | **0.96** | **0.92** | **0.94** | **0.05** | **0.95** |
| CTGAN-based manipulation | AdGraph [21] | 0.74 | 0.78 | 0.85 | 0.81 | 0.26 | 0.75 |
| | WebGraph [37] | 0.89 | 0.88 | 0.87 | 0.87 | 0.11 | 0.96 |
| | **AdFlush** | **0.98** | **0.98** | **0.97** | **0.98** | **0.02** | **0.98** |

## 5.4 Runtime Overhead Evaluation

To evaluate AdFlush's runtime overhead, we measured its time complexity on a Ubuntu 18.04 with an Intel Xeon E5-2687W v3 CPU and 256GB of memory within a Python environment. We calculated the average time from model loading to test dataset inference over 10 trials. AdFlush's inference time of 2.3 seconds was significantly faster than the state-of-the-art solutions (17.4 seconds for AdGraph [21], 19.7 seconds for WebGraph [37], and 144.9 seconds for WTAGraph [43]), marking an 86% speedup over AdGraph. This faster inference speed underscores AdFlush's efficacy for real-time ad and tracker blocking.

We also evaluated AdFlush's performance on the top 1,000 Tranco websites using Selenium to replicate user engagements on a Chrome browser. Typical interactions were conducted via page scrolling and a 60-second timeout. We collected 68,495 queries from the 1,000 sites. For a single request, AdFlush averagely took 0.242 seconds (SD = 0.990 seconds) for feature extraction and 0.014 seconds (SD = 0.035 seconds) for inference, respectively. It is important to note that these tasks do not take time in an additive manner. Utilizing JavaScript's asynchronous functionalities, AdFlush performs simultaneously without disrupting the user's experience and is much speedier than results from the user's perspective.

AdFlush is also superior in terms of CPU and memory consumption. For CPU usage, AdGraph and WebGraph consumed 1.8% and 1.6%, respectively, while AdFlush used only 0.8%, requiring 56% less CPU than AdGraph. WTAGraph also reported 0.8% CPU usage, but its memory demand was significantly higher than AdFlush's. WTAGraph required a substantial 1.3 GB of memory during inference, while AdGraph and WebGraph needed over 200 MB and 150 MB, respectively. In comparison, AdFlush's memory footprint was a modest 40 MB, a reduction of up to 80% from AdGraph.

## 6 DISCUSSION

**Validity of Ground Truth Dataset.** Machine learning models that detect ads and web trackers, such as AdFlush, often rely on filter lists like EasyList [9] as their ground truth. The accuracy of these models is inherently linked to the reliability of these lists. Sole reliance on filter lists introduces two primary challenges: (1) The need for frequent updates to ensure that the lists are comprehensive and up-to-date, and (2) potential inaccuracies in the lists that could jeopardize the model's trustworthiness. Le *et al.* [26] highlighted these issues and proposed AdHighlighter as a solution; however, this tool also requires regular maintenance. As described in Section 4.1, to mitigate these challenges, we used eight popular filter lists (detailed in Appendix C) to obtain valid labels for our collected dataset. This approach helped reduce the risk of bias from relying on a single filter list.

**Feasible Features for Real-Time Ad Blocking in Browser Extensions.** State-of-the-art ML-based ad and web tracker detectors often emphasize model accuracy without thoroughly considering the feasibility of implementing their features in real-world web browser extensions. For ad-blocking extensions, the primary objective is to halt requests intended for ads and web trackers in real-time. However, many graph-based features used in existing ML-based solutions require the outcomes of HTTP requests and the execution results of JavaScript code, indicating that the ad and tracker code must be executed before detection occurs. To address this challenge, we exclusively considered features that can be swiftly computed in real-time within a browser extension. This allows AdFlush to quickly generate rules for blocking URLs in real-time before HTTP requests are delivered and code is executed.

## 7 CONCLUSION

We have introduced AdFlush, a novel framework for effectively detecting ads and trackers. AdFlush stands out with its carefully selected features, which surpass those of state-of-the-art tools. In addition to the 350 features used by state-of-the-art tools, we consider 533 new JavaScript features. We evaluated the efficacy of all 883 features and chose a subset of 27 optimal features for AdFlush, considering both feature importance and feasibility of browser implementation.

On a benchmark of the top 10,000 websites, AdFlush achieved an F1 score of 0.98, outperforming the best state-of-the-art tool, AdGraph [21], by 0.05 in F1 score. AdFlush also demonstrated remarkable resilience against adversarial attacks, surpassing state-of-the-art methods. Additionally, AdFlush maintained its high detection accuracy for six months without retraining, demonstrating its effectiveness for new and unseen data.

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

## A CATEGORIZATION OF CANDIDATE FEATURES

Table 5 shows the features that we investigate in this paper, including all aggregated features from AdGraph [21], WTAGraph [43], and WebGraph [37], as well as new features against JavaScript obfuscation strategies. Duplicate features are grouped by implementation, and we group the entire feature set into JavaScript, URL, HTTP header, and graph categories based on their resource. Specific features considered for AdFlush are marked with "‡" in the table.

**Table 5: Summary of the categorized features. Feature sets include AdGraph (A), WTAGraph (T), and WebGraph (W), with AdFlush (‡) denoting new features. A '●' indicates a tool's use of the feature, while '✓' signifies the feasibility (F) of integrating the feature into a browser extension.**

| Theme | Feature | A | T | W | F |
|---|---|---|---|---|---|
| JavaScript | # of requests sent | | ● | ● | ✓ |
| | # of storage (set/get) | | ● | ● | ✓ |
| | # of cookies (set/get) | | ● | ● | ✓ |
| | Use of *eval/function* instructions | ● | ● | ● | ✓ |
| | Use of ad/tracker related JavaScript API | | ● | | ✓ |
| | *n*-gram frequency of JavaScript AST‡ | | | | ✓ |
| | JavaScript AST depth/breadth‡ | | | | ✓ |
| | Average of *Identifier* length‡ | | | | ✓ |
| | Average of characters per line‡ | | | | ✓ |
| | Bracket to dot notations ratio in JavaScript‡ | | | | ✓ |
| URL | URL length | ● | ● | | ✓ |
| | Sub-domain check | ● | | | ✓ |
| | Valid query string parameters | ● | | | ✓ |
| | Domain party (first/third) | ● | | | ✓ |
| | Specific keywords in request URL | ● | | | ✓ |
| | Semi-colons in request URL | ● | | | ✓ |
| | Base domain in request URL | ● | | | ✓ |
| | Screen size in request URL | ● | | | ✓ |
| | Character embeddings | | | ● | ✓ |
| HTTP header | Content policy type (categorical) | ● | ● | | ✓ |
| | Content policy type (boolean) | | | ● | ✓ |
| | Request Content type (boolean) | | | ● | ✓ |
| | Request Method type (boolean) | | | ● | ✓ |
| | Order, timing, and # of attributes | | | ● | ✓ |
| Graph | # of nodes/edges and node/edge ratio | ● | | ● | |
| | Degree (in, out, in+out) | ● | | ● | |
| | Average degree connectivity | ● | | ● | |
| | Presence of ancestor script | ● | | ● | |
| | Presence of parent script | ● | | ● | ✓ |
| | Ascendant script length | ● | | ● | |
| | Ad keywords in ascendant script | ● | | ● | |
| | Use of *eval/function* in ascendant | ● | | ● | |
| | Use of *eval/function* in descendant | ● | | ● | |
| | # of ancestors/descendants | | | ● | |
| | Closeness centrality/eccentricity | | | ● | |
| | # of predecessors/successors script | | | ● | |
| | # of requests received | | | ● | |
| | # of redirects sent/received | | | ● | |
| | Max depth of redirect | | | ● | |
| | # of indirect degree (in/out) | | | ● | |
| | # of indirect ancestors/descendants | | | ● | |
| | Indirect centrality/connectivity | | | ● | |
| | Indirect in/out weights (min/max/mean) | | | ● | |
| | # of set then get/modify storage (src/dst) | | | ● | |
| | # of set/get URL (src/dst) | | | ● | |
| | # of all indirect degree (in/out) | | | ● | |
| | # of all indirect ancestors/descendants | | | ● | |
| | Indirect all centrality/connectivity | | | ● | |

Section 3.3 explores the feasibility of implementing each feature within a privacy-preserving browser environment. Following our rules to ensure users' privacy, we note whether the feature is secure to implement in a commercial web browser. Only one feature in the Graph category is available by observing the `Initiator` in the request header without referencing other requests or response headers. For the same reason, the *max_depth_of_redirect* is only observable by chaining requests and responses with a communicated server. Therefore, extracting this feature is vulnerable to ad and tracker collecting user privacy.

## B FEATURE SET OF ADFLUSH

AdFlush uses a carefully selected collection of the most effective features to detect ads and web trackers. These features are detailed in Table 6 and are used in both Python and web browser environments. For example, we compute the number of requests sent, storage values set or fetched, and accesses of cookies based on the occurrences of related JavaScript API calls, similar to how WebGraph does it. We also calculate the average length of *Identifiers* by averaging the length of names parsed as *Identifier* tokens within JavaScript source code and HTML <script> tags. Finally, we use a Boolean feature to indicate whether a unique character follows an ad or tracker keyword within the request URL.

**Table 6: Features used in AdFlush. The types stand for Numerical (N), Boolean (B), and Categorical (C).**

| Category | Feature | Type | # of Values |
|---|---|---|---|
| JavaScript | # of requests sent | N | 1 |
| | # of storage set | N | 1 |
| | # of storage get | N | 1 |
| | # of cookies get | N | 1 |
| | *n*-gram frequency of JavaScript AST | N | 6 |
| | Average of *Identifier* length | N | 1 |
| | Average of characters per line | N | 1 |
| | Bracket to dot notation ratio in JavaScript | N | 1 |
| URL | Third-party check | B | 1 |
| | Request URL length | N | 1 |
| | Ad keyword with special character in URL | B | 1 |
| | FQDN character embeddings | N | 7 |
| | Request URL character embeddings | N | 3 |
| HTTP header | Content policy type | C | 1 |
| **Total** | - | - | **27** |

Overall, AdFlush uses a variety of features to detect ads and web trackers. These features are carefully selected to be effective and efficient, and they are implemented in a way that preserves user privacy.

## C DATASET LABELING WITH FILTER LISTS

To label our dataset, we aggregated continually updated filter lists, which served as our ground truth dataset. We collected the filter lists on April 4, 2023, using the same methodology as Iqbal *et al.* [21] and Siby *et al.* [37]. However, Anti-Adblock Killer, Blockzilla, and Squid Blacklist were no longer actively updated or supported, so we incorporated two additional filter lists, uBlock Origin, and uBlock Privacy, to bolster our ground truth dataset. Table 7 shows the lists used in AdFlush, along with the number of rules in each list as of the collection date and the source of each filter list. By integrating

filter lists beyond those specified in Table 7, AdFlush's performance can potentially improve due to more reliable labels.

**Table 7: Filter lists for labeling.**

| Filter List (# Rules) | URL |
|---|---|
| EasyList [9] (53,997) | https://easylist.to/easylist/easylist.txt |
| EasyPrivacy [10] (30,646) | https://easylist.to/easylist/easyprivacy.txt |
| Fanboy annoyance [13] (4,644) | https://easylist.to/easylist/fanboy-annoyance.txt |
| Fanboy social [13] (18,883) | https://easylist.to/easylist/fanboy-social.txt |
| Peterlowe [31] (3,667) | https://pgl.yoyo.org/adservers/serverlist.php?hostformat=adblockplus |
| uBlock Origin [18] (11,152) | https://raw.githubusercontent.com/uBlockOrigin/uAssets/master/filters/filters.txt |
| uBlock Privacy [18] (219) | https://raw.githubusercontent.com/uBlockOrigin/uAssets/master/filters/privacy.txt |
| Warning removal list [41] (697) | https://easylist-downloads.adblockplus.org/antiadblockfilters.txt |

## D HTTP REQUEST COVERAGE OF ADFLUSH

Table 8 provides a detailed breakdown of AdFlush's detection performance across 14 types of HTTP requests in our Top-10K dataset. AdFlush captures all 14 request types, as outlined in Google Chrome's documentation.

As the table shows, AdFlush identifies a similar number of HTTP requests as the filter lists for each request type. Overall, AdFlush and filter lists detect 39.27% and 39.92% of all requests as advertisements and web trackers, respectively, demonstrating that AdFlush can effectively serve as a substitute for filter lists.

Notably, AdFlush demonstrates exceptional proficiency across the majority of request categories. For instance, it achieves an accuracy, precision, recall, and F1 score exceeding 0.99 for the three most common request types, `image`, `script`, and `XMLHttpRequest`, which collectively constitute 80.41% of all dataset requests.

However, AdFlush's performance is somewhat diminished for `imageset`, `main_frame`, and `other` request types, with false negative rates of 0.06, 0.13, and 0.09, respectively. These request types have a skewed distribution of true and false values, with only 1.75%, 3.60%, and 3.30% true values per the filter lists.

## E ADS AND TRACKERS MISSED BY FILTER LISTS

In our longitudinal study detailed in Section 5.2, we identified significant samples of advertisements and trackers that were mislabeled by even commercial filter lists, but that AdFlush could detect. We manually analyzed and verified 642 such URLs from 108 unique domains to discern their unique types and behaviors.

We found that `iframe` is frequently used for ads and trackers. These `iframe` tags contained several code snippets designed to execute JavaScript segmented into elements. An example is shown in Figure 6, where the `eval()` function is used to process inner HTML text as source codes within alternative containers.

AdFlush also detected `other` requests to JavaScript resources that were later added to conventional filter lists. Figure 7 shows an example of a detected tracker source code that transmits valuable user data to a server using `sendBeacon`. Notably, AdFlush was able to accurately detect this tracker code, which is not detected by commercial filter lists.

**Figure 6: Script detected by AdFlush that the filter lists missed.**

**Figure 7: Example tracker code detected by AdFlush.**

**Figure 8: Examples of tracker behavior in URL and HTTP headers.**

Table 8: Detection performance of AdFlush by HTTP request type.

| Request Type | # Requests | # Blocked by Filter Lists | # Detected by AdFlush | Accuracy | Precision | Recall | F$_1$ | FNR | FPR |
|---|---|---|---|---|---|---|---|---|---|
| image | 312,231 | 120,120 | 119,422 | 0.99 | 0.99 | 0.99 | 0.99 | 0.01 | 0.00 |
| script | 237,509 | 112,772 | 111,961 | 0.99 | 0.99 | 0.99 | 0.99 | 0.01 | 0.00 |
| xmlhttprequest | 117,826 | 59,528 | 59,139 | 0.99 | 0.99 | 0.99 | 0.99 | 0.01 | 0.00 |
| stylesheet | 43,674 | 3,449 | 3,358 | 0.99 | 0.99 | 0.97 | 0.99 | 0.03 | 0.00 |
| font | 33,314 | 3,642 | 3,626 | 0.99 | 0.99 | 0.99 | 0.99 | 0.00 | 0.00 |
| imageset | 26,382 | 462 | 434 | 0.99 | 0.99 | 0.94 | 0.97 | 0.06 | 0.00 |
| sub_frame | 25,620 | 19,014 | 18,985 | 0.99 | 0.99 | 0.99 | 0.99 | 0.00 | 0.01 |
| main_frame | 18,459 | 665 | 573 | 0.99 | 0.99 | 0.86 | 0.93 | 0.13 | 0.00 |
| ping | 11,736 | 11,231 | 11,218 | 0.99 | 0.99 | 0.99 | 0.99 | 0.00 | 0.02 |
| media | 1,654 | 281 | 278 | 0.99 | 0.99 | 0.98 | 0.99 | 0.02 | 0.00 |
| other | 696 | 23 | 21 | 0.99 | 0.99 | 0.91 | 0.95 | 0.09 | 0.00 |
| websocket | 511 | 132 | 128 | 0.99 | 0.99 | 0.97 | 0.98 | 0.03 | 0.01 |
| csp_report | 398 | 58 | 54 | 0.99 | 0.98 | 0.93 | 0.96 | 0.07 | 0.00 |
| object | 150 | 12 | 12 | 0.99 | 0.99 | 0.99 | 0.99 | 0.00 | 0.00 |
| **Total** | 830,160 | 331,389 | 329,205 | 0.99 | 0.99 | 0.99 | 0.99 | 0.01 | 0.00 |

Finally, we present examples of tracker behavior evident in URLs and HTTP headers. Request ID 114,851, illustrated in Figure 8, is directed at an image URL but returns a single pixel. This tactic is used to transmit user data to the tracker through the query string. Similarly, Request ID 59,437, another image URL, responds with a single pixel due to the storage of user identifiers in the browser cookie. Although filter lists overlooked these instances of privacy breaches, AdFlush efficiently mitigated them.

To further explore filter lists' false negatives, we relabeled our dataset with the latest filter lists from October 11, 2023. By manually verifying the newly labeled AdFlush detections, we obtained 642 URLs from 108 unique domains where all authors agreed on the verification. We reported these results to EasyList, EasyPrivacy, uBlock Origin, and uBlock Privacy to help create more effective and updated filter lists.

