# OpenReview forum: "AdFlush: A Real-World Deployable Machine Learning Solution for Effective Advertisement and Web Tracker Prevention"
_ACM.org/TheWebConf/2024/Conference — TheWebConf24_

### Official Review · Reviewer_UFnp · 2023-11-17

**Novelty:** 4
**Technical Quality:** 4

**Review:**

The authors have done a very thorough piece of work in which they create a machine learning based algorithm that uses 27 features and removes (effectively detects) ads. They are of course limited by Google Manifest, however, they do the best that can be done. They use different machine learning algorithms, mostly based on boosting. The most interesting part of the study is the longitudinal study over a period of 6 months. This piece of work is reasonably solid and the comparison with related work is quite sound. The authors claim to have compared with three state-of-the-art works. The results are also favorable.

My review scores are still not stellar because as a problem, this is very old. Using ML algorithms, it is always possible to do a lot of fine tuning and tweaking to make it more robust and also get a good accuracy. I would have ideally loved to see a new kind of ML technique being used such as transformers or diffusion-based models (not sure about their relevance for this problem, though). Something new could have definitely made this paper shine. Also, there was no need to send so much of data to a third-party server (requestly.io). Can something be done more locally?

**Questions:**

1. Can we say something definitive about state-of-the-art work? What makes them inferior?
2. Were more sophisticated ML models considered? Do the boosting based approaches considered by the authors the best?
3. Is there an element of explainability in the model?

**Reviewer Confidence:**

3: The reviewer is confident but not certain that the evaluation is correct

**Scope:**

4: The work is relevant to the Web and to the track, and is of broad interest to the community

---

### Official Review · Reviewer_c4mY · 2023-11-22

**Novelty:** 5
**Technical Quality:** 6

**Review:**

The paper presents a lightweight ML-based ad blocker that outperforms state-of-the-art solutions in both effectiveness (F1 score) and computation.

I appreciate the contributions and presentation of the work done, and would be happy to see the paper accepted at the conference. The practical angle makes the work all the more valuable. The paper is also written very clearly.

The paper makes a convincing case that it outperforms previous ML-based ad blocking methods. The finding that despite making the approach more lightweight, the model is able to also achieve a better detection rate, was very interesting and somewhat counter intuitive to me. I do not know how feasible it is to go deeper into why this is the case (e.g., providing an intuition for the feature set in Appendix B), but it is intriguing nonetheless.

There are quite a few deeper and comprehensive insights that I find very valuable, for example the feature distributions (Figure 2), longitudinal study, and robustness analysis.

I greatly appreciate that there will be a publicly available Chrome extension, and that the authors did the effort to report the undetected ad URLs to the main filter list providers.

I see a few main areas for improving the paper.

 The first is to provide evidence for certain statements that are relevant to the contributions but are not (sufficiently) substantiated:
* S1: _"AdFlush does not transmit User-Agent values or cookies to third parties, upholding a strong commitment to user privacy"_ - who does this then?
* S2.3: _"automated filter rule generation is inherently unsuitable
for real-time ads and trackers detection" - why?
* S3.4: _"Additionally, obfuscation-related features, such
as Identifier lengths in ASTs and character counts in code lines,
would also enhance performance."_ - any proof for this?
* S4.2: _"We
analyzed the training data for each feature selection, identified the
top 10 models based on F1 scores, and selected the best-performing
model using 5-fold cross-validation"_ - can you give the results? It is unclear whether the GBM vastly outperforms the other models, or whether their performances are very similar instead.

The second is showing more concretely where the newly developed features (Sections 3.2, 3.4) stem from, and whether they were solicited systematically. The paper does not go in much detail beyond "we identified new features"; the only (short) explanation I found is in Section 3.2: _"These new features are derived
from requested JavaScript source code or embedded HTML scripts,
including n-gram frequencies parsed from the abstract syntax tree
(AST), the structure of the AST, and various script-based metrics."_
I would value a more detailed description of how these features were developed and selected.

The third is addressing the impact of two error-related issues:
* Figure 4: there is a fixed delay timeout of 300ms. What would happen if the evaluation takes longer than 300ms? Will the ad then be shown to the user? In this area, I was also confused by the fact that in S5.4, the standard deviation for a single request is 0.990s, bringing the total request time well above 300ms.
* What is the impact of false positives/negatives on the user experience, also per HTTP request type? Some broad insights would be helpful. For example, does it matter that there is a FNR of 13% for `main_frame` requests? Does this mean up to 13% of ads can be shown to the user? Or for the FPR of 2% for `ping`, are potentially necessary requests blocked, and could this break website functionality?

For the Runtime Overhead Evaluation (S5.4), I am confused by the use of such a high-resource machine (Intel Xeon CPU, 256 GB RAM) for evaluating what is claimed to be a "lightweight" approach. The results suggest that AdFlush is more and rather lightweight (e.g., 40 MB RAM usage), but I do not see why this should be evaluated on a non-consumer device. Some more detail on whether the evaluation results apply in total, to one request, to one web page, ... would also be very helpful. This feels necessary to me to clarify, as the computational performance is one of the main contributions claimed.

A final comment on the extension (S4.3). I wonder what the impact is of already having to resort to an older version of `declarativeNetRequest`, could this mean that the extension could very quickly stop working if such version would be fully deprecated or discouraged from using. (This comment does not affect my evaluation of the paper in any way.)

Minor nitpicks:
- S1. Introduction: _"To address these issues several tools are available, ..."_: missing a "such as"?
- S3.1: an average of 32.58 seconds per crawl gives me 32.58 s * 10000 sites / 7 crawlers = 13 hours, yet 11 hours are mentioned
- Appendix A (Table 5) could benefit from the "# of values" column that is already in Table 6 - this would also resolve the "confusion" touched upon in footnote 1.

The reasoning for my novelty score is that the area of ML-based ad blocking has had quite a body of research already (so the problem/solution category is less "novel"), but the paper still contributes greatly to improving the state of the art.

**I have read the rebuttal.**

**Questions:**

- Explain the procedure behind soliciting and selecting the newly developed features.
- Clarify the impact of the 300ms timeout, and the reason why a single request could take (much) longer to execute.

**Reviewer Confidence:**

3: The reviewer is confident but not certain that the evaluation is correct

**Scope:**

3: The work is somewhat relevant to the Web and to the track, and is of narrow interest to a sub-community

---

### Official Review · Reviewer_xnxa · 2023-11-22

**Novelty:** 4
**Technical Quality:** 5

**Review:**

Pros:
- The paper demonstrates practical improvements to many issues facing ML ad blockers.
- The problem is well motivated, since manual lists are easy for advertisers to bypass.
- The implementation uses clever ways to operate within the limits of the Manifest V3 requirements.

Cons:
- Claims sometimes stronger than evidence provided in the paper


My main review is about toning down the strength of some claims. For example:
"Manual maintenance of these filter lists requires significant human effort and they are prone to false-positive and false-negative errors."  Machine learning models also arguably require human effort to curate. The model requires human effort to collect ground truth data, and periodically retrain with new data. Furthermore, the paper argues that the ML model is more robust against adversarial examples. The paper supports the claim that AdFlush is more robust to adversarial examples than other ML methods, but not that it is more robust than filter lists. I think more acknowledgment of these things in a limitations section would help.

The TPR at FP rate 0 is a more relevant metric for this situation than F1. For a practical deployment in the web scenario, controlling false positives is very important to users since a false positive can break websites. Therefore, for this application, I would argue that the most relevant metric would be the true positive rate at a FP rate close to zero (like somewhere between 10e-4 to 10e-7).

For performance evaluation, it would be more relevant to report the latency impact on loading a whole page. The metrics per request look promising, but the user experience is more closely linked to the entire latency impact on the page. A comparison to existing latency for filter lists would also be good here.

**Questions:**

Questions:
- What are the model's metric for TPR at a FPR close to 0?
- Of the 642 additional detections by the model, how many were not manually verified?
- How does latency compare to existing filter list implementations?
- How does latency affect the entire load time of the page?

**Reviewer Confidence:**

3: The reviewer is confident but not certain that the evaluation is correct

**Scope:**

4: The work is relevant to the Web and to the track, and is of broad interest to the community

---

### Official Review · Reviewer_rrga · 2023-11-30

**Novelty:** 5
**Technical Quality:** 5

**Review:**

Firstly, I would like to thank the authors for submitting their paper to WWW'24! It was definitely interesting to read and addresses a highly relevant and timely topic (ad/tracker blocking), and focuses on the practicality of deploying such a countermeasure. Overall, the paper demonstrates an improved performance, both in terms of detection ability as well as resource usage. Furthermore, I appreciated the longitudinal study showing that the feature drift over time is limited and consequently the drop in performance is acceptable.

Nevertheless, there are a couple of concerns with the study that I believe should be addressed:

- The study operates under the premise that advertisers/trackers are not trying to actively circumvent detection. Assuming that the model is published for all users (which would be required to run the extension on the users' browsers), the advertiser/tracker could keep modifying the scripts and setup until they are able to bypass the detection. As long as the recall is not 1, such cases can exist.
- Given that only 27 features are considered, it would have been interesting to see an overview of how difficult it would be for an advertiser/tracker to modify such features. For instance changing the number of storage gets to just a single one seems like mostly a development issue.
- The key feature identification was performed separately for existing and new features, and then those 47 and 11 features were just combined. This makes the assumption that the combination does not affect their performance.

**Questions:**

- Which features are intrinsic to the advertising/tracking ecosystem, i.e., which features would be extremely difficult to manipulate or adjust without changing the way the ecosystem works?
- Why were the new and existing features considered separately for the key feature identification?

**Ethics Review Description:**

No ethical concerns.

**Reviewer Confidence:**

3: The reviewer is confident but not certain that the evaluation is correct

**Scope:**

4: The work is relevant to the Web and to the track, and is of broad interest to the community

---

### Official Review · Reviewer_iKtG · 2023-12-02

**Novelty:** 4
**Technical Quality:** 4

**Review:**

Thank you for submitting your paper to WWW 2024. I do like the direction that you take in your work, but I am unconvinced about its efficacy and whether it is actually practical. This makes it difficult for me to support acceptance. The main problem I see is that you do not compare or evaluate your approach to non-ML ad blockers in either performance or accuracy. Instead, your approach tests whether you can detect some parts of the existing filter lists, but it is not clear which ones and what their reach is. Moreover, considering especially approaches like ad proxying, CNAME cloaking, and the recent tricks YouTube has been playing, blocking at the request-level may be insufficient for current advertisements (and some filter lists go way beyond specifying just request URLs, for example, uBlock Origin's lists, which are very different from EasyList filter lists).

Therefore, I find it is necessary to also evaluate against "traditional" ad blocks that are actually the state of the art in ad blocking (especially uBlock Origin), rather than focusing on ML-based ad blockers and research prototypes that have not been adopted or used widely. Indeed, many of the "traditional" ad blockers do not only block at the request level based on filter lists (like EasyList), but go beyond those, rendering a comparison at this level inaccurate and superficial.

Overall, I would have loved to see an actual end-to-end evaluation of your ad blocker on a portion of some top website list (e.g., Tranco) against existing traditional ad blockers and ML-based blockers, investigating how many ads you actually block and also what the performance overhead is. Currently, you do these investigations separately, without considering the environment your ad blocker runs in, which I do not find convincing.

**Questions:**

- If you would deploy AdFlush in a browser, how do website load times change? Prior work at WWW 2020 has shown that privacy-focused browser extensions can reduce load times, and it would be useful to understand how AdFlush compares to them.
- What are the ads that AdFlush cannot detect? Do they follow any pattern, or do they fall into specific groups?

**Reviewer Confidence:**

3: The reviewer is confident but not certain that the evaluation is correct

**Scope:**

3: The work is somewhat relevant to the Web and to the track, and is of narrow interest to a sub-community

---

### Decision · Program_Chairs · 2024-01-22

**Decision:**

Accept

**Comment:**

This paper presents AdFlush, a lightweight ML-based ad/tracker detection system. To develop AdFlush, the authors first identify and analyze hundreds of features, and then condense them into 27 fearues that are used to build a classifier. AdFlush compares favorably against state-of-the-art systems graph-based ad detection systems, such as AdGraph, WebGraph, and WTAgraph. During the discussion phase, the authors also provided a comparison with traditional ad blockers such as uBlock Origin. Besides extensive comparisons against previous work in terms of accuracy and performance, the paper also includes an analysis of potential evasions, showing that AdFlush is more robust to adversarial manipulations, compared to previous work. Additionally, the paper also includes a longitudinal study of AdFlush's performance, showing that AdFlush remains effective over time.

 Overall, the reviewers seem to appreciate the work, though they had some requests for improvement. For instance they asked for a comparison with traditional ad blockers and evaluation results that are more centered around reducing false positives, rather than simply measuring the F1 score. Also, they asked for a more in-depth analysis of the features. The authors responded well to the reviewers' questions and presented convincing evidence to support the effectivencess of AdFlush as a practical ad blocker.

 ---